# Encapsulation of Metronidazole in Biocompatible Macrocycles and Structural Characterization of Its Nano Spray-Dried Nanostructured Composite

**DOI:** 10.3390/molecules26237335

**Published:** 2021-12-02

**Authors:** Mirella Mirankó, Mónika Megyesi, Zsombor Miskolczy, Judit Tóth, Tivadar Feczkó, László Biczók

**Affiliations:** 1Research Institute of Biomolecular and Chemical Engineering, Faculty of Engineering, University of Pannonia, Egyetem St. 10, 8200 Veszprém, Hungary; miranko@mukki.richem.hu (M.M.); toth@mukki.richem.hu (J.T.); 2Research Centre for Natural Sciences, Institute of Materials and Environmental Chemistry, Eötvös Loránd Research Network (ELKH), P.O. Box 286, 1519 Budapest, Hungary; megyesi.monika@ttk.hu (M.M.); miskolczy.zsombor@ttk.hu (Z.M.)

**Keywords:** drug delivery, self-assembly, host–guest binding, inclusion complex, spray drying, 4-sulfonatocalix[4]arene, cucurbit[7]uril, chitosan, metronidazole

## Abstract

Due to the great potential of biocompatible cucurbit[7]uril (CB7) and 4-sulfonatocalix[4]arene (SCX4) macrocycles in drug delivery, the confinement of the pharmaceutically important metronidazole as an ionizable model drug has been systematically studied in these cavitands. Absorption and fluorescence spectroscopic measurements gave 1.9 × 10^5^ M^−1^ and 1.0 × 10^4^ M^−1^ as the association constants of the protonated metronidazole inclusion in CB7 and SCX4, whereas the unprotonated guests had values more than one order of magnitude lower, respectively. The preferential binding of the protonated metronidazole resulted in 1.91 pH unit pK_a_ diminution upon encapsulation in CB7, but the complexation with SCX4 led to a pK_a_ decrease of only 0.82 pH unit. The produced protonated metronidazole–SCX4 complex induced nanoparticle formation with protonated chitosan by supramolecular crosslinking of the polysaccharide chains. The properties of the aqueous nanoparticle solutions and the micron-sized solid composite produced therefrom by nano spray drying were unraveled. The results of the present work may find application in the rational design of tailor-made self-assembled drug carrier systems.

## 1. Introduction

The formulation of active pharmaceutical ingredients is a significant challenge in the development of stable and bioavailable dosage forms with improved therapeutic efficacy. Water-soluble macrocycles possessing a large hydrophobic interior, e.g., cyclodextrins, calixarenes and cucurbiturils, are widely used as functional excipients because the inclusion of drugs in their cavity can lead to enhanced solubility, a modified release rate, prolonged chemical or thermal stability, improved delivery and reduced side effects [1,2,3]. The host–guest interaction with macrocyclic cavitands has been exploited to create versatile nanoscale drug-delivery systems [4], biomedical applications [5] and to overcome antibiotic resistance [6]. The self-organization of such assemblies has the advantages of easy preparation and facile tuning of the properties. The dynamic nature of noncovalent interactions allows the fabrication of stimuli-responsive nanostructures for a targeted and controlled release of the biologically active constituent [7,8,9].

These potential benefits of host–guest complex formation may be exploited in the formulation of ionizable model drug, metronidazole (Met), a widely used synthetic antibiotic and antiprotozoal agent [10]. Various approaches have been reported to enhance its bioactivity and poor water-solubility, but its encapsulation has not been studied in cucurbit[7]uril (CB7) and 4-sulfonatocalix[4]arene macrocycles (SCX4). Met was included in mucoadhesive tablets composed of cellulose and polyacrylic acid derivatives [11], nanospheres made of amphiphilic β-cyclodextrin [12] and in calcium pectinate beads [13]. Cyclodextrins and mesoporous silica particles were combined as a composite to reduce the rate of drug liberation [14]. An ionic gelation technique was employed to create Met-loaded chitosan nanoparticles capable of pH-triggered sustained release [15,16]. Met was also incorporated in nanofibrous webs by electrospinning [17,18,19], whereas crystalline–amorphous nanostructured micronized solid dispersions were prepared by the spray drying of solutions containing polymers [20]. Met induced a micelle formation of poly(1-vinyl-2-pyrrolidone) and the linear polymer showed the best loading capacity [21].

Despite the various attempts to improve the formulation of Met, the marketed products containing this pharmaceutically active compound still have many side effects. Hence, it is worth exploring new alternative drug carrier systems. We shall reveal how the experimental conditions affect the binding affinity of Met to cucurbit[7]uril and 4-sulfonatocalix[4]arene biocompatible cavitands. Our previous results demonstrated that the larger homologue of SCX4, 4-sulfonatocalix[8]arene, comprising eight sulfonatophenol units, not only served as a supramolecular crosslinking agent promoting the association with protonated chitosan to nanoparticles but also provided binding sites for inclusion complex formation with pharmaceutically important alkaloid molecules [22]. As an extension of this work, we also examine the encapsulation of metronidazole in nanoparticles prepared by the self-assembly of SCX4 and chitosan in acidic aqueous solution. Since spray drying is a promising technology in modern pharmaceutical industries [23], and is one of the processes used for the stabilization of nanosuspensions, we elucidate how this procedure affects the characteristics of the self-assembled chitosan–MetH^+^–SCX4 nanoparticles. Figure 1 displays the structural formula of the compounds used.

## 2. Results and Discussion

### 2.1. Effect of CB7 on the pH Dependence of the Absorption Spectra

The broad lowest-energy absorption band of Met has a maximum at 320 nm in a neutral aqueous solution. A less intense maximum appears at 227 nm and absorbance grows below 220 nm (Appendix A). The absorption maximum shifts to 277 nm and a hypochromicity of the band is observed in 0.1 N H_2_SO_4_ solution due to the protonation of the nucleophilic unsubstituted nitrogen of the imidazole ring.

The triangles in Figure 1a present the absorbance (*A*) alteration at 320 nm with pH, whereas the lines display the results of the nonlinear least-squares analysis by the following relationship:(1)$$A=\frac{A_{0}-A_{\infty }}{1+\exp[\left(pH-pK_{a}\right)\ln10]}+A_{\infty }$$
where *A*_0_ and *A_∞_* are the absorbances at low and high pHs. The best fit of Equation (1) to the experimental data provided *pK_a_* = 2.36 for the negative logarithm of the equilibrium constant of the proton dissociation, in excellent agreement with the previously published *pK_a_* = 2.38 value [24]. From the calculated *A_0_* and *A_∞_*, *ε*_1_ = 8730 M^−1^ cm^−1^ and *ε*_2_ = 780 M^−1^ cm^−1^ were obtained for the molar absorption coefficient of Met and its protonated form (MetH^+^) at 320 nm.

The spectral changes occur at a much higher pH in 500 μM CB7 aqueous solution, indicating inclusion complex formation. The absorption spectra barely differ in the presence and absence of 500 μM CB7, implying that the nitro-substituted part of the heterocyclic moiety is located outside the host cavity. The absorption characteristics remain unchanged in CB7 solution when the pH is reduced from alkaline to the value of 5.75 and resemble those of the unprotonated Met. In more acidic solutions, the gradual absorbance decrease of this band is accompanied with the emergence of a new band with a peak at 275 nm (Figure 1b), which is attributed to the host–guest complex of the protonated metronidazole (MetH^+^) and CB7. Isosbestic points appear at 282, 251 and 221 nm, demonstrating that it is only a 1:1 association that occurs. Spectral alterations vanish below pH 3 since further protonation cannot take place. The nonlinear least-squares fit of the pH-dependent absorbances gave *pK_a_^′^* = 4.29 for the negative logarithm of the equilibrium constant of the proton dissociation from the MetH^+^–CB7 inclusion complex. Figure 1a shows that the encapsulation in CB7 brings about a 1.91 pH unit displacement of the sigmoid titration curve toward a less acidic range for the protonation equilibrium of Met. The acidity diminution upon embedment in CB7 originates from the stabilization of the MetH^+^ form by cation–dipole interactions with the partial negative charge of the carbonyl-laced portals of the macrocycle. Host–guest binding induced proton release suppression was also found for various dyes, drugs and ammonium compounds [25,26,27,28,29].

When the smaller homologue, cucurbit[6]uril (CB6), composed of 6 glycoluril units linked by a pair of methylene groups, was used as a host, no evidence was found for MetH^+^ complexation. The absorption spectra of the Met or MetH^+^ solutions were not modified in the presence of CB6. The interior of CB6 seems to be too small to encapsulate these guests.

### 2.2. Determination of the Binding Constant of Inclusion in CB7

Met inclusion in CB7 was studied at pH 7.5 because it remained unprotonated both in water and in the CB7 cavity under this condition (Figure 1a). The addition of cavitand brought about a minor change in the absorption spectrum. Therefore, dehydrocorydaline was applied as a fluorescent probe for the determination of binding affinity. Appendix A displays the variation of the fluorescence spectra in the aqueous solution of 1.0 μM dehydrocorydaline and 3.0 μM CB7 upon the gradual addition of Met at pH 7.5. Due to the competitive binding of Met, fluorescence intensity is reduced because the fluorescence quantum yield of the probe is much smaller in water than in CB7 [30]. The experimental data were analysed as described in our previous paper [31], and 2.0 × 10^6^ M^−1^ was taken for the equilibrium constant of dehydrocorydaline inclusion in CB7 [28]. The line in the inset to Appendix A represents the result of the nonlinear least-squares fit corresponding to a *K* = 1000 M^−1^ binding constant for Met–CB7 formation.

The equilibrium constant of MetH^+^ confinement in CB7 was determined at pH 3 because most of the Met–CB7 complex is protonated under this condition (see Figure 1a). The equimolar (412 μM) solution of host and guest was gradually diluted and the change of the absorption spectrum was monitored. The apparent molar absorption coefficients (*ε*), calculated using the total metronidazole concentrations, are plotted as a function of wavelength in Figure 2a.

A substantial rise of the characteristic broad band with a maximum at 320 nm was found, indicating the dissociation of MetH^+^–CB7. The exit of MetH^+^ from the host macrocycle was followed by rapid proton loss, resulting in Met and only a small fraction of the released MetH^+^ remained protonated at pH 3 (see Figure 1a). The shape of the initial spectrum recorded for the most concentrated solution differs from that displayed in Figure 1b for a similar pH. The displacement of the maximum to 280 nm and higher absorption above 300 nm suggest that not only MetH^+^–CB7 and a small amount of Met–CB7 was produced but also unbound Met, MetH^+^ and CB7 remained in the solution. The protonation of CB7 can be neglected because its equilibrium constant is only ~166 M^−1^ [32]. The dependence of the apparent molar absorption coefficients (*ε*) on the equimolar total concentration of the solutes ([*Met*]*_T_* = [*CB7*]*_T_*) (Figure 2b) is expressed by:(2)$$\varepsilon =\varepsilon_{1}\frac{\left[Met\right]}{{\left[Met\right]}_{T}}+\varepsilon_{2}\frac{\left[MetH^{+}\right]}{{\left[Met\right]}_{T}}+\varepsilon_{3}\frac{\left[Met–CB7\right]}{{\left[Met\right]}_{T}}+\varepsilon_{4}\frac{\left[MetH^{+}–CB7\right]}{{\left[Met\right]}_{T}}$$
where *ε*_1_, *ε*_2_, *ε*_3_ and *ε*_4_ represent the molar absorption coefficients of Met, MetH^+^, Met–CB7 and MetH^+^–CB7, respectively, and the concentrations are:(3)$$\left[Met\right]=\frac{-q+\sqrt{q^{2}+4\left(K+K_{p}\left[H^{+}\right]/K_{a}\right)q{\left[CB7\right]}_{T}}}{2\left(K+K_{p}\left[H^{+}\right]/K_{a}\right)q}$$
where $q=\left(1+\frac{\left[H^{+}\right]}{K_{a}}\right)=1.2291$, while *K*, *K_p_* and *K_a_* represent the equilibrium constants of Met–CB7, MetH^+^–CB7 formations and MetH^+^ deprotonation, respectively.
(4)$$\left[MetH^{+}\right]=\frac{\left[Met\right]\left[H^{+}\right]}{K_{a}}$$
(5)$$\left[Met–CB7\right]=K\left(1+\frac{\left[H^{+}\right]}{K_{a}}\right){\left[Met\right]}^{2}$$
(6)$$\left[MetH^{+}–CB7\right]=K_{p}\frac{\left[H^{+}\right]}{K_{a}}\left(1+\frac{\left[H^{+}\right]}{K_{a}}\right){\left[Met\right]}^{2}$$

The Appendix A shows how these equations were obtained. The equilibrium constant of MetH^+^ deprotonation (*K_a_* = 4.365 × 10^−3^ M) was derived from the *pK_a_* = 2.36 value determined by spectrophotometric titration (vide supra). The acidity of the solution was kept constant ([*H*^+^] = 10^−3^ M). The *ε*_1_ = 8730 M^−1^ cm^−1^, *ε*_2_ = 780 M^−1^ cm^−1^ and *K* = 1000 M^−1^ values were the results of independent experiments (vide supra). The best fit of the experimental results with Equations (2)–(6) provided *K_p_* = 1.9 × 10^5^ M^−1^, *ε*_3_ = 8800 M^−1^ cm^−1^ and *ε*_4_ = 620 M^−1^ cm^−1^. The line in Figure 2b presents the calculated function. A previous study demonstrated that the complexes of CB7 can cross the cell membrane [33]. Hence, the encapsulation in CB7 is expected to facilitate the transport of metronidazole into cells.

### 2.3. Association with SCX4

The complex formation of MetH^+^ was also examined with the polyanionic flexible SCX4 host compound. Appendix A shows that this macrocycle does not absorb light above 325 nm, irrespective of pH. The titration of Met solution with sulphuric acid in the presence of 2 mM SCX4 leads to absorbance change in a less acidic pH range than in neat water (Figure 3), indicating complex formation. The global fit of the data in the 325–400 nm spectral domain with Equation (1) provided *pK_a_′* = 3.20 for the negative logarithm of the equilibrium constant of the deprotonation of MetH^+^–SCX4. The diminution of the MetH^+^ acidity by 0.82 pH units upon the complexation with SCX4 arises from the preferential binding of the cationic protonated guest to the negatively charged cavitand.

The association constant of MetH^+^–SCX4 formation was determined at pH 2, where the sulfonic acid moieties of SCX4 are completely deprotonated but the phenolic OH groups do not lose protons because their first dissociation step has a *pK_a_* of 3.2 [34]. The macrocycle has four negative charges in such circumstances. The small difference between the absorption spectra of free and SCX4-complexed MetH^+^ did not allow the attainment of accurate information on binding affinity by a spectrophotometric method. Therefore, berberine was used as a fluorescent probe. Our previous study has shown that the environment sensitive emission of this plant alkaloid can be exploited for the investigation of the competitive binding of optically silent guests to 4-sulfonatocalix[n]arenes (SCXn) in an acidic aqueous solution [31,35].

Figure 4 presents the variation of the fluorescence spectra and the emission intensity at 570 nm in 5.0 μM berberine (B) and 243 μM SCX4 aqueous solution at pH 2 when the total metronidazole concentration is progressively raised. The gradual expulsion of B from SCX4 results in fluorescence weakening. Taking into account the competitive binding of B, Met and MetH^+^ to SCX4, the experimental data were analysed as described in the Appendix A, using 3720 M^−1^ for the equilibrium constant of B–SCX4 complex formation [35]. The line in the inset to Figure 4 represents the best nonlinear least-squares fit, which provides 1.0 × 10^4^ M^−1^ for MetH^+^ association with SCX4. The equilibrium constant of Met–SCX4 complex formation was found to be <1000 M^−1^. The more than one order of magnitude difference in the association constants of MetH^+^ and Met can be applied in the design of pH-responsive drug delivery systems. Such behaviour has not been reported when cyclodextrins serve as hosts for Met [12,14].

### 2.4. Nanoparticle Formation of Chitosan Induced by the MetH^+^-SCX4 Complex

To demonstrate that SCX4 can serve not only as a drug carrier but that it also promotes the supramolecular crosslinking of protonated chitosan polysaccharide chains, the self-assembly of chitosan and SCX4 was studied in the presence and absence of 20 mM metronidazole at pH 2. The repeat unit concentration of protonated chitosan (2.07 mM) was kept constant and the amount of SCX4 was varied. Under our experimental conditions, the phenolic OH moieties of SCX4 and the NH_2_ groups of chitosan were fully protonated. Due to the four negative charges of the former compound and the 84% degree of deacetylation of the polymer (vide infra in the Experimental section), the macrocycle:polymer charge ratio (Q) was derived as follows: Q = (4(SCX4))/(0.84(chitosan)). The gradual rise of Q led to significant turbidity enhancement (Figure 5A) due to nanoparticle formation. Coagulation occurred in the absence of MetH^+^ when Q approached 1, but large aggregates are not created even at Q = 1.1 macrocycle:polymer charge ratio in solutions containing MetH^+^. These results indicate the incorporation of MetH^+^ in the nanoparticles.

After the embedment of a MetH+ cation in SCX4, the complex produced becomes triply charged. Hence, more complexed SCX4 macrocycles can be included in the nanoparticles before the positive charge of protonated chitosan is counterbalanced compared to the amount of free SCX4 necessary for charge neutralization.

To reveal why the turbidity was always lower in the presence of MetH^+^, dynamic light scattering experiments were performed. The self-organization resulted in smaller particles with a narrower size distribution in solutions comprising MetH^+^, causing lower turbidity. The mean diameter of these nanoparticles varied to a lesser extent with the growth of the SCX4 amount compared to that found without MetH^+^ (Figure 5B). Figure 5D displays representative size distributions. Based on the binding constant of MetH^+^–SCX4 formation (*K* = 1.0 × 10^4^ M^−1^, vide supra), it can be calculated that practically all SCX4 macrocycles were complexed by MetH^+^ under our experimental conditions. Thus, MetH^+^–SCX4 complexes served as noncovalent crosslinkers connecting the polymer chains within the nanoparticles. Zeta potential measurements (Figure 5C) showed that positively charged nanoparticles were produced in accordance with the excess of the NH_3_^+^ moieties of the polymer over the negative charges of the MetH^+^–SCX4 complexes. At low SCX4–MetH^+^ contents, a zeta potential of 39 mV was observed, which remained unchanged until the embedment of further SCX4–MetH^+^ complexes altered only the inner core of the nanoparticles. As the amount of SCX4–MetH^+^ came close to the particle charge neutralization, the zeta potential gradually vanished. Our approach to inducing the nanoparticle formation of protonated chitosan by SCX4–MetH^+^ complexation has the advantage of providing smaller particle diameters (d = 168–302 nm) than the previously published optimised method (d = 558 nm) that employed a tripolyphosphate supramolecular crosslinker for metronidazole embedment [15]. The smaller carrier particles usually improve the bioavailability of the drugs.

### 2.5. The Effect of Spray Drying the Nanoparticle Solutions

Drying is a useful process for the stabilization of nanosuspensions. The prepared nanosuspension (solution (S1) of DS1 sample) was encapsulated in a hydroxypropyl methylcellulose (HPMC) polymer matrix in order to produce powder by a nano spray-drying process. The use of the polymer was necessary because drying without it produced very sticky Met, which could not be extracted from the device. For the sake of comparison, solid samples were also prepared by nano spray drying the 0.01 M HCl solutions containing Met + HPMC and only HPMC (DS2 and DS3, respectively). The yield of the processes, the active ingredient content of solutions and solid products, particle diameter and size distribution data are given in Table 1.

The Met content was preserved during the drying process, although the process yield was only 60.2 and 46.5% for DS1 and DS2, respectively. The average particle size of DS1 was smaller than that of DS2 (see D(4,3) in Table 1). The size distributions of the two samples were different (see Figure 6): the DS1 sample had a bimodal distribution, while in the case of DS2 the two peaks merged.

The SEM image of the DS1 sample (Figure 7A) shows that agglomerated particles of polymer spheres and angular crystals dominate. When the DS2 sample composed of HPMC and Met was examined, well-developed Met crystals appeared in the SEM image together with the polymer spheres (Figure 7B). The morphology of the dried polymer (the DS3 sample) is presented in Figure 7C.

The ingredients (Figure 8) and solid products (Figure 9) were also characterized by differential scanning calorimetry (DSC). The locations of the maxima of the calorimetric peaks in the 100–200 °C temperature domain are summarized in Table 2. The Met melting point at 161.0 °C (159–163 °C in [36]) appeared with a sharp peak. SCX4 had an endotherm peak at 124.3 °C, which can be assigned to the water loss of the molecules [37]. HPMC and chitosan formed amorphous states. The DSC curves of the dried samples showed several endotherm peaks belonging to the melting process of crystalline phases.

The DSC curves of the DS1 and DS2 samples (Figure 9) were very similar concerning the endotherm peaks, which can be attributed to the melting of MTZ. The crystals of MTZ can be identified in the SEM images in Figure 7A,B. Double peaks appeared at lower temperatures (see Table 2, Peak1) than the melting point of Met, i.e. 161.0 °C. Multiple melting peaks can be attributed to two different crystals or morphologies [38]. Melting point depression can be explained by the presence of impurities or the reduction of crystal size. In our previous paper [20], this phenomenon was discussed in spray-dried composites of HPMC and Met. The melting point decrease was attributed to the reduction of crystal size; the first endotherm signal corresponded to the melting of smaller crystals, while the second larger endotherm peak was the result of the melting of the larger crystal fraction.

The third, higher melting peak (see Table 2, Peak 2) at 166.2–168.3 °C can probably be attributed to the presence of HCl as a different chemical environment for the ionizable Met molecules. It is difficult to obtain clear information on the state of the Met in the composites with a higher ratio of polymers. The nano spray drying of pure Met in HCl solution did not yield a collectable product; thus, a solution (S4) of the DS4 sample was prepared with a higher ratio of Met. Based on the evaluation of the recorded infrared spectra of DS4 (see Figure 10), there were some changes observed in comparing the active ingredient and composite spectra. The imidazole ring vibration at 1581 cm^−1^ was shifted to 1610 cm^−1^ and a shoulder appeared at 1549 cm^−1^, which is a shift of the N–O stretching vibration from 1534 cm^−1^. These changes in the composite infrared spectrum compared with the samples containing the pure Met or the polymer HPMC only provide evidence that some of the active ingredients slightly changed the molecular structure due to the formation of secondary bonds (e.g., van der Waals or hydrogen bonding) in the composite.

In the dried product, the drug can be dispersed macroscopically in a crystalline state or molecularly in an amorphous state. The amorphous form is more effective for solubility enhancement, while the crystalline form has a better chemical and physical stability [39]. At least to some extent, the powder XRD images justified the crystalline state of the metronidazole in the composites (see Figure 11). It can clearly be seen from the images that the peak intensities of Met differ between the DS1 and DS2 samples (although there is a minimal difference in drug content), suggesting a higher content of the amorphous form of the drug in the DS1 than in the DS2 sample. This is probably due to the embedment of Met in calixarene–chitosan nanoparticles, thereby inhibiting the crystallization of Met.

The spray-drying process results in a storable and concentrated form of the drug-carrier system, which is an important benefit. Moreover, to our knowledge, calixarene complexes have not been spray dried as yet. In our nano spray dried chitosan–MetH^+^–SCX4 product, the drug has been found to be present partly in a stable crystalline and partly in an amorphous form.

## 3. Materials and Methods

Metronidazole (Met) was kindly provided by Egis Pharmaceuticals PLC, Hungary. Cucurbit[7]uril (Sigma-Aldrich, St. Louis, MO, USA), cucurbit[6]uril (Sigma-Aldrich) and 4-sulfonatocalix[4]arene (SCX4) (Acros Organics, Thermo Fisher Scientific, Geel, Belgium) were dried in a high vacuum prior to use. Berberine chloride (Sigma-Aldrich, St. Louis, MO, USA) was chromatographed on a silica gel (Merck, Budapest, Hungary) column eluted with ethanol. Dehydrocorydaline (BOC Sciences, Shirley, NY, USA) was used as received. Low molecular weight chitosan (Fluka, Honeywell GmbH, Hannover, Germany) had an 84% degree of deacetylation based on the ^1^H NMR method developed by Hirai and coworkers [40]. The amount of the dissolved polymer was given as the concentration of the saccharide repeat unit for which a molecular mass of 168.56 Da was employed. Hydroxypropyl methylcellulose (HPMC) was purchased from Colorcon Ltd. Water was distilled twice from a dilute KMnO_4_ solution. Experiments were performed at 296 K.

The pH of the aqueous solutions, adjusted with HCl or NaOH, was measured with a Consort C832 apparatus, the glass electrode of which was calibrated with buffer standards. Nanoparticles were prepared by mixing the required volumes of protonated chitosan and SCX4 stock solutions of pH 2 under stirring. Both solutions contained 0.02 M metronidazole when we intended to incorporate it in the nanoparticles. Absorption spectra were recorded on an Agilent Technologies Cary60 spectrophotometer, whereas corrected fluorescence spectra were taken on a Jobin-Yvon Fluoromax-4 photon counting spectrofluorometer. The turbidity (T) values were obtained from the absorbances (A) at 450 nm by the relationship T = 1–10^−A^. Dynamic light scattering and zeta potential measurements were carried out on a Zetasizer Nano-ZS (Malvern Instrument, Malvern, UK) equipped with a He–Ne laser (λ = 633 nm, scattering angle 173°) after a 1 h equilibration time. The reported mean diameters are calculated based on intensity distribution.

### 3.1. Nano Spray Drying Experiment

The drying experiment was carried out using a Nano Spray Dryer B-90 (Büchi Labortechnik AG, Flawil, Switzerland). The basic principle of the operation of the dryer is detailed elsewhere [41]. The preheated gas enters at the top of the chamber. Droplet generation is based on vibration mesh technology, i.e., an electronically driven piezoelectric actuator vibrates the thin spray mesh. The addition rate is adjusted by means of a recirculation pump and spray rate. The droplets solidify via solvent evaporation and the electrostatic collector collects the dried particles from the leaving gas.

The drying gas inlet temperature was 100 °C, with a flow rate of 90 L/min, the mesh of the membrane was 7 µm and the recirculation pump rate and the spray rate were 60% and 45%, respectively. The drying experiment was carried out using the tall set-up employed for water-based samples. For drying experiments, nanosuspensions or solutions were prepared as follows:-S1: 0.750 m/m% HPMC polymer was dissolved in a suspension of 0.352 m/m% Met, 0.030 m/m% SCX4 and 0.034 m/m% chitosan prepared in 0.01 M HCl (CMet,i. = 30.19 m/m%, where CMet,i. is the Met concentration in the nanosuspension).-S2: 0.750 m/m% HPMC polymer was dissolved in solutions of 0.352 m/m% Met prepared in 0.01 M HCl, (CMet,i. = 31.94 m/m%).-S3: 0.750 m/m% HPMC was dissolved in 0.01 M HCl solution.-S4: 0.100 m/m% Met and 0.100 m/m% HPMC were dissolved in 0.01 M HCl solution (CMet,i. = 50.00 m/m%).

### 3.2. Active Ingredient Content

The Met content in the dried samples was measured by spectrophotometry. For the investigations, a Shimadzu UV-1800 spectrophotometer (Shimadzu, Kyoto, Japan) was used. For the calibration curve, standard solutions were prepared in distilled water as follows: 0.025 g of Met was dissolved in 100 mL distilled water. 5 mL of this solution was diluted to 25 mL and further dilutions were made to concentrations of 2.5, 5, 10, 20, 30 and 40 µg/mL. The absorbance at 319 nm was measured. For the spray-dried sample, 20 mg product was dissolved in distilled water and diluted to a concentration of 80 µg/mL. After dissolution, the chitosan was centrifuged by Hermle Z216 MK centrifuge (15,000 rpm, 20 min, 20 °C, Hermle AG, Gosheim, Germany). The Met content of the dried sample was calculated using the following Equation (7):(7)$$C_{Met}=\frac{\left(A_{319}+0.0019\right)/0.0518}{80}\times 100$$
where *C_Met_* = Met content (m/m %) in the dried sample, A_319_ = absorbance at λ = 319 nm.

### 3.3. Particle Size and Distribution

The particle size and distribution were determined by laser diffraction. For the calculation of the size distribution, a refractive index of 1.5 was used and the imaginary component of the refractive index was 0.5. The particle size was reported as a volume mean diameter, marked as D(4,3), and the size distribution with d10, d50 and d90 data. The measurements were carried out in wet dispersion in 100 mL of cyclohexane containing 0.1 m/m% soy lecithin with a Malvern Mastersizer 2000 (Malvern Instruments, Malvern, UK) using the SM dispersion unit with a stirring rate of 2000 rpm. A sample of 10 mg was taken in 1 mL of cyclohexane solutions containing 0.1 m/m% soy lecithin and sonicated for 40 s at 30% of power with a 6 mm probe using a Sonics VCX 130 (Sonics & Materials, Newtown, CT, USA), after which the obtained suspension was loaded onto the dispersion unit for measurement.

### 3.4. Morphology

The surface morphology of the samples was examined using an FEI Thermofisher Apreo S (Thermo Fisher Scientific, Waltham, MA, USA) scanning electron microscope. During the investigation, 5 kV accelerating voltage was used.

### 3.5. X-ray Diffraction and FTIR Measurements

X-ray diffraction images were recorded using a Philips PW 3710 diffractometer (Philips Analyical, Almelo, Netherlands) with CuKα radiation, a tube current of 40 mA and a voltage of 50 kV at a scanning rate of 0.02° 2 θ/s. Control of the device and data collection was achieved with Philips X’Pert Data Collector software. The FTIR spectra were recorded with a Varian Scimitar FTS2000 spectrometer (64 scans, 4 cm^−1^ resolution) equipped with a liquid nitrogen-cooled MCT detector and a Pike GladiATR (with diamond micro-ATR element) accessory.

### 3.6. Thermal Measurements

Thermal measurements were performed on a Setaram LabsysEvo (Lyon, France) TG-DSC system, in flowing high purity argon (99.999%, flow rate 50 mL/min) atmosphere. Samples were weighed into 100 μL aluminium crucibles (the reference cell was empty) and were heated from 25 °C to 300 °C with a heating rate of 10 °C/min. The obtained data were blank corrected and further processed with the thermoanalyser’s processing software (Calisto Processing, v2.092). The thermal analyser (both the temperature scale and calorimetric sensitivity) was calibrated by a multipoint calibration method, in which seven different certified reference materials were used to cover the thermal analyser’s entire operating temperature range.

## 4. Conclusions

Both CB7 and SCX4 macrocycles can serve as molecular containers for the encapsulation of the protonated form of the medically important metronidazole. The less efficient encapsulation of the unprotonated metronidazole compared to the protonated variety in both hosts can be exploited to secure pH-responsive drug release. Due to their multianionic character, MetH^+^–SCX4 complexes can self-assemble with the polycationic protonated chitosan polymer into drug-loaded nanoparticles of ~200 nm mean diameter. Spray drying of these nanoparticle suspensions proved to be an effective method to produce a micron-sized solid composite incorporating metronidazole. In the solid dispersion obtained, Met was integrated in a polymer matrix in a crystalline–amorphous phase according to the DSC and XRD analysis. The amorphous Met content was significantly higher in the product dried from the SCX4–chitosan nanoparticle suspension than in that spray-dried from the metronidazole + HPMC solution—a result attributable to the electrostatic repulsion between MetH^+^–SCX4 complexes in acidic media.

## Data Availability

Data are contained within the article.

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
