# Peer review of "Encapsulation of Metronidazole in Biocompatible Macrocycles and Structural Characterization of Its Nano Spray-Dried Nanostructured Composite"

_molecules, 2021, doi:10.3390/molecules26237335_

Round 1

Reviewer 1 Report

The authors here studied the delivery of metronidazole (Met) using biocompatible cucurbit [7]uril (CB7) and 4-15 sulfonatocalix[4]arene (SCX4) cavitands and envisioned a pH responsive drug release system. MetH+−SCX4 complexes self-assembled with the polycationic protonated chitosan polymer into drug-loaded nanoparticles and spray drying of these nanoparticle solutions produced micron-sized solid composite incorporating metronidazole.

Overall, the manuscript looks good. However, there are a few comments which should be addressed before publication. I recommend a major revision of the manuscript before publication.

  1. First and foremost, what is the advantage of using this system? The authors themselves have mentioned in the Introduction that plenty of techniques were utilized to improve the bioactivity and poor water-solubility of Met. What are the new facets that this system offers compared to the pre-existing ones? This should be clearly explained in the manuscript, both in a discussion and results sections.
  2. What is the basis of choosing Met since many research has already been done with this drug?

Minor comment:

  1. The English should be improved. Some sentences are very difficult to comprehend.
  2. Abstract, line 6, sentence should be shortened.

Author Response

We thank Reviewers for the comments and have made the following changes in line with the suggestions:

Reviewer 1

  1. First and foremost, what is the advantage of using this system? The authors themselves have mentioned in the Introduction that plenty of techniques were utilized to improve the bioactivity and poor water-solubility of Met. What are the new facets that this system offers compared to the pre-existing ones? This should be clearly explained in the manuscript, both in a discussion and results sections.

Response

The advantage has been emphasized in the introduction:

“These potential benefits of host-guest complex formation may be exploited in the formulation of ionizable model drug, metronidazole, a widely used synthetic antibiotic and antiprotozoal agent [10]. Various approaches have been reported to enhance its bioactivity and poor water-solubility but its encapsulation has not been studied in cucurbit[7]uril (CB7) and 4-sulfonatocalix[4]arene macrocycles (SCX4). “ …

“Despite the various attempts to improve the formulation of Met, the marketed products containing this pharmaceutically active compound still have many side effects. Hence, it is worth exploring new alternative drug carrier systems. Now, we shall reveal how the experimental conditions affect the binding.”

Reviewer 1

  1. What is the basis of choosing Met since many research has already been done with this drug.

Response

One reason was that metronidazole is an ionizable model drug. Further, our industrial partner, EGIS Pharmaceuticals suggested its application in this study as a model drug.

Reviewer 1

Minor comment:

  1. The English should be improved. Some sentences are very difficult to comprehend.

Response

The text of the manuscript has been revised by a native English speaker.

  1. Abstract, line 6, sentence should be shortened.

Response

Abstract, line 6: The sentence has been modified as follows:

“Although weaker binding was observed to SCX4 than to CB7, practically complete drug incorporation could be achieved even in the former cavitand in acidic solution. The produced protonated metronidazole-SCX4 complex induced nanoparticle formation with protonated chitosan by supramolecular crosslinking of the polysaccharide chains.”

Reviewer 2 Report

In their work, the authors report in detail the structural characterization of spray-dried micronized nanocomplexes of metronidazole (Met, “guest”) -macrocyclic cavitands [“hosts”:

 cucurbit[7]uril (CB7), 4-sulfonatocalix[4]arene (SCX4)]. Characterization methods were: spectrophotometric / fluorometric analysis, SEM, X-ray diffraction, FTIR and DSC. The study presents relevant information, yet certain changes may improve the scientific soundness and uniqueness of this study:

  • The readability and syntax of the manuscript will be substantially improved if it is reviewed by a formal translation agency or by a native English spoken person.
  • Title. It must be precise and related to the content. Suggestion: "Structural characterization of spray-dried nanostructured metronidazole-macrocyclic cavitands"
  • Abstract. It should be more concise and quantitative without sacrificing important differential results. CB7 and SCx4-based complexes related differences must be highlighted
  • Introduction & conclusion. Should be shorter and more focused.
  • Results & discussion. Any methodological description should be removed from this section and sent to the respective section (e.g. lines 100-107; “The results…at 320 nm”).
  • Tables (T) & Figures (F). Please improve the resolution of almost all figures (300 dpi or more) and all figures must be extensively explained, otherwise include as supplementary material. Tables require a better format according to Molecules
  • Methods. Methods included as supplementary material should be part of the main body of the text. Specific analytical conditions and other relevant technical considerations must be included in each section.
  • References. OK

Author Response

Reviewer 2

  1. Title. It must be precise and related to the content. Suggestion: "Structural characterization of spray-dried nanostructured metronidazole-macrocyclic cavitands".

Response

We thank the reviewer for the comment. The results on the 1:1 host-guest binding to CB7 and SCX4, as well as the nanoparticle formation initiated by metronidazole-SCX4 complex are important new findings, which we prefer to mention in the title to attract the attention of the members of wide scientific community. Thus, we changed the second part of title to emphasize the structural characterization of spray-dried nanoparticles.

Reviewer 2

  1. Abstract. It should be more concise and quantitative without sacrificing important differential results. CB7 and SCx4-based complexes related differences must be highlighted.

Response

The abstract has been revised as Reviewer suggested. To make it more quantitative, association constants and pKa change values were included for both CB7- and SCX4-based complexes.

Reviewer 2

  1. Introduction & conclusion. Should be shorter and more focused.

Response

The last paragraph of the introduction and lines 3-8 of the conclusion have been deleted.

Reviewer 2

  1. Results & discussion. Any methodological description should be removed from this section and sent to the respective section (e.g. lines 100-107; “The results…at 320 nm”)..

Response

The text has been modified to explain the results presented in Figure 1a instead of providing methodological description.

Reviewer 2

  1. Tables (T) & Figures (F). Please improve the resolution of almost all figures (300 dpi or more) and all figures must be extensively explained, otherwise include as supplementary material. Tables require a better format according to Molecules.

Response                                   

The figures are prepared with 600 dpi resolution and the tables have been formatted following the journal guidelines. Some more explanations have been added to the text about some figures (e.g. Fig. 10)

Reviewer 2

  1. Methods. Methods included as supplementary material should be part of the main body of the text. Specific analytical conditions and other relevant technical considerations must be included in each section.

 Response

The supplementary material has been corrected accordingly.

Round 2

Reviewer 1 Report

The main question that still remains is the novelty of the work. Although the authors have tried to include the advantage of the system in the introduction section, it is still not convincing. The result section should contain comparisons with the other systems reported demonstrating advantages of the mentioned one.

Author Response

Reviewer 1:

"The main question that still remains is the novelty of the work. Although the authors have tried to include the advantage of the system in the introduction section, it is still not convincing. The result section should contain comparisons with the other systems reported demonstrating advantages of the mentioned one."

Response:

We completed the discussion with the following important information regarding the novelty:

 A previous study demonstrated that the complexes of CB7 can cross the cell membrane [33]. Hence, the encapsulation in CB7 is expected to facilitate the transport of metronidazole into cells.

 ...

The more than one order of magnitude difference in the association constants of MetH+ and Met can be applied in the design of pH-responsive drug delivery systems. When cyclodextrins serve as hosts for Met [12,14], such a behaviour has not been reported.

 ...

Our approach to induce nanoparticle formation of protonated chitosan by SCX4-MetH+ complex has the advantage of providing smaller particle diameters (d = 302 – 168 nm) than the previously published optimised method (d = 558 nm) that employed tripolyphosphate supramolecular crosslinker for metronidazole embedment [15]. The smaller carrier particles usually improve the bioavailability of the drugs.

...

The spray drying process results in a storable and concentrated form of the drug-carrier system, which is an important benefit. Moreover, to our knowledge, calixarene complexes has not been spray dried, yet. In our nano spray dried chitosan-MetH+-SCX4 product, the drug has been present partly in a stable crystalline and partly in amorphous form.